# Water-Soluble Dicationic Deuteroporphyrin Derivative for Antimicrobial PDT: Singlet Oxygen Generation, Passive Carrier Interaction and Nosocomial Bacterial Strains Photoinactivation

**Dmitry B. Berezin** [1,*]**, Sergey O. Kruchin** [2]**, Natal'ya V. Kukushkina** [1,2]**, Evgeny A. Venediktov** [2]**,**
**Mikhail O. Koifman** [1] and **Andrey V. Kustov** [1,2]

1   Research Institute of Macroheterocyclic Compounds, Ivanovo State University of Chemistry and Technology (ISUCT), 153012 Ivanovo, Russia
2   G.A. Krestov Institute of Solution Chemistry, Russian Academy of Sciences (ISC RAS), 153045 Ivanovo, Russia
*   Correspondence: berezin@isuct.ru

**Abstract:** Multidrug resistance of pathogenic microflora is a serious threat to the modern community looking for new approaches to treating superinfections. In this sense, antimicrobial photodynamic therapy (aPDT) is an effective and safe technique considered to be a promising alternative or an important supplement to the traditional clinically applied methods for inactivating antibiotic resistant pathogens. Macroheterocyclic photosensitizers (PS) of three generations are proposed for clinical practice. They are known as the key compounds for PDT able to be localized selectively in microbial cells and to be activated with the red light producing toxic reactive oxygen species (ROS). However, these neutral and anionic PSs possess low affinity towards the outer lipopolysaccharide membrane of Gram-negative bacteria and, consequently, poor ability to kill these pathogens under irradiation. In contrast, cationic PSs containing one or more charged groups, especially those bound to an appropriate carrier, provide efficient inactivation of microorganisms. In this paper, we focus on the study of photophysics, aggregation and photoinduced antimicrobial activity of the water-soluble derivative of deuteroporphyrin-IX, a blood group porphyrin, bearing two cationic trialkylammonium fragments. This potential photosensitizing agent is found to generate singlet oxygen in a non-polar environment and forms stable nano-sized molecular complexes with passive non-ionic carrier Tween 80, localizing in an aqueous surfactant solution as a non-aggregated form in the surface micellar layer. Two different modes of PS/Tween 80 binding characterized by their own stability constants and interaction stoichiometry are observed. Microbiological experiments clearly demonstrate that the increased permeability of the outer bacterial membrane caused by the application of the intramicellar form of the photosensitizer or addition of some potentiation agents leads to pronounced light phototoxicity of the pigment against antibiotic-resistant nosocomial strains of Gram-negative bacterial  pathogens.

**Keywords:** antimicrobial photodynamic therapy; antibiotic resistant microorganisms; dicationic porphyrin photosensitizer; synthesis; water solubility; singlet oxygen generation; PS–Tween 80 molecular complexation; photoinduced antibacterial activity

## 1. Introduction

Resistance of the bacterial pathogens from the ESCAPE group to all known classes of antibiotics and further expansion of nosocomial strains non-sensitive to antimicrobial drugs in the last few years is an alarming threat to the modern community [1–3]. In this situation, antimicrobial photodynamic therapy (aPDT) can be considered a promising alternative way to treat superficial and other localized infections of different origins [2–7]. The simple idea behind aPDT is irradiation of the organic dye (PS), intensively absorbing in the therapeutic window of living tissues (620–820 nm) and located in the site of infection, with a visible red light in the presence of $O_2$ molecules to make use of the killing effect of photo-generated reactive oxygen species (ROS) [2–12]. This non-invasive low-toxic therapeutic

technique is used to inactivate, in vitro and in vivo, both antibiotic-resistant microorganisms [3–5,11,13–16] and cancer cells [6–8,10,17–20]. Macroheterocyclic photosensitizers of three generations medically approved for clinical practice are represented by amphiphilic porphyrin, chlorin or phthalocyanine molecules substituted with functional groups of neutral ("Photofrin", "Foscan", etc.) or anionic ("Photosens", "Fotolon", "Talaporfin", etc.) nature [8,10,13,14,17,19,21–26]. At the same time, the presence of cationic groups in the PS structure not only provides appropriate water solubility of the photosensitizing agents and dramatically changes their affinity towards cell membranes and mechanisms of PS biodistribution [5,7,20,27–29], but also strongly enhances the efficacy of the photodynamic inactivation of Gram-negative pathogens during the aPDT compared with neutral and negatively charged molecules [3–5,11,13–15,20].

Our scientific group is involved in an extensive and continuing series of multidisciplinary studies on the development and investigation of the known and new potential sensitizers for antitumor and antimicrobial PDT based on different macroheterocyclic platforms, including PSs with cationic groups [7,12,14–16,18–20,30]. For instance, the chlorin photosensitizer bearing one trialkylammonium substituent demonstrated good solubility in water at physiological temperatures and a good quantum yield of singlet oxygen generation in a lipid-like environment, high affinity to the blood lipoproteins predicting selective PS accumulation in tumor cells, as well as powerful photoinduced antimicrobial and antitumor activity at both low dark cellular and acute toxicity of the drug [14,20,31,32]. Here, we present our experimental results on the solubility, partition, singlet oxygen generation, interaction with the non-ionic passive carrier Tween 80, and photoinactivation of antibiotic-resistant bacteria by a water-soluble dicationic blood group porphyrin derivative (Compound **5**) newly synthesized (Scheme 1, Comp. **1**–**5**) from hemin (Comp. **1**) [33]. The experimental material provided includes a comparative analysis of the behavior of DPD, dicationic chlorophyll *a* derivative (Comp. **6**) studied by us earlier [14,34] and the well-established PS "Fotolon" based on trianionic chlorin $e_6$ (Comp. **7**, Figure 1) [7,18–20,25,35,36].

**Scheme 1.** Synthesis of 13(3),17(3)-*bis*-N-(2-N′,N′,N′-trimethylammonioethyliodide) amide of deuteroporphyrin-IX (Comp. **5**) from hemin (Comp. **1**).

**Figure 1.** Structural formula of 3(1),3(2)-*bis*-(N,N,N-trimethylammoniomethyliodide)-13(1)-N′-methylamide-15(2),17(3)-dimethyl ester of chlorin e$_6$ (Comp. **6**) and chlorin e$_6$ trisodium salt (Comp. **7**).

## 2. Materials and Methods

### 2.1. Chemicals

Protohemin-IX (purity >99%) was purchased from "AppliChem". Resorcinol (Reachem, Moscow, Russia, >99.5%), diethyl ether ((C$_2$H$_5$)$_2$O, Vekton, Saint Petersburg, Russia, >99%), iron (II) sulfate heptahydrate (FeSO$_4$ × 7H$_2$O, AquaChem, Ivanovo, Russia, 99%), chloroform (CHCl$_3$, Vekton, Saint Petersburg, Russia, >98%), methanol (CH$_3$OH, Vekton, Saint Petersburg, Russia, >99%), acetyl chloride (AcCl, Geel, Belgium, Acros Organics, 98%), hydrochloric acid (HCl, EKOS-1, Staraya Kupavna, Russia, 35%), acetone ((CH$_3$)$_2$CO), Vekton, Saint Petersburg, Russia, >99%), acetic acid (AcOH, Geel, Belgium, Acros Organics, 98%), alumina oxide (Al$_2$O$_3$, Aldrich, Buchs, Switzerland, >99%), N,N-dimethylethylenediamine (Acros Organics, Geel, Belgium, >99%), carbon tetrachloride (CCl$_4$, Vekton, Saint Petersburg, Russia, >99%), methyl iodide (CH$_3$I, Aldrich, Taufkirchen, Germany, >99.5%), benzene (Vekton, Saint Petersburg, Russia, >99%), tetraphenylporphyrin (H$_2$TPP, Geel, Belgium, Acros Organics, >0.98%), 1-octanol (OctOH, Panreac, Barcelona, Spain, >0.99%), hexane (Vekton, Saint Petersburg, Russia, >98%), ethanol (EtOH, Vekton, Saint Petersburg, Russia, >96%), sodium ethylenediaminetetraacetate (Na$_2$H$_2$Edta × 2H$_2$O, Panreac, Barcelona, Spain, chemical pure), sorbitan monooleate (Tween 80, Panreac, Barcelona, Spain, pure, pharma grade), potassium iodide (KI, MosChemTorg, Moscow, Russia, >99%), hydrogen peroxide (H$_2$O$_2$, 50% aqueous solution, Aldrich, Taufkirchen, Germany), silica (pore size 60 Å, 63–200 µm particle size, Aldrich, Buchs, Switzerland) were used as supplied. Water was distilled twice.

### 2.2. Synthesis of PSs

The target compound (Comp. **5**) was prepared by the four-stage synthesis procedure presented in Scheme. The starting 13(3),17(3)-dimethyl ester of deuteroporphyrin-IX (Comp. **3**) was obtained from natural hemin (Comp. **1**) using the Schumm devinylation reaction [37] followed by reductive demetalation and esterification of the macroheterocycle [38]. The water-soluble derivative of PS **3**, 13(3),17(3)-*bis*-N-(2-N′,N′,N′-trimethylammoniaethyl iodide) amide of deuteroporphyrin-IX (Comp. **5**), was synthesized by a two-stage synthesis, including aminolysis with N,N-dimethylethylenediamine and further quaternization of the two tertiary amino groups of Comp. **4** formed in the reaction with methyl iodide (Scheme). The yield of compound **5** from PS **3** was 70%.

Dicationic chlorin, 3(1),3(2)-*bis*-(N,N,N-trimethylammoniomethyl iodide)-13(1)-N′-methylamide-15(2),17(3)-dimethyl ester of chlorin e$_6$ (Comp. **6**), was synthesized as recommended in paper [39].

13(3),17(3)-Dimethyl ester of deuteroporphyrin-IX. At the first stage of the synthesis, 1 g of protohemin-IX (Comp. **1**, 1.54 mmol) was triturated with 3 g of resorcinol (C$_6$H$_6$O$_2$) and the mixture was alloyed for 1 h at 453 K. After cooling, the intermediate was extracted with diethyl ether until a colorless extract was obtained. Then, the solvent was evaporated, and the solid residue of deuterohemin-IX (Comp. **2**, Scheme) was dried. At the next stage,

1 g (1.62 mmol) of the deuterohemin-IX powder (Comp. **2**) and 2.5 g of FeSO$_4$ × 7H$_2$O were suspended in a mixture of 150 mL of methanol and 150 mL of chloroform. Next, 7.5 mL of acetyl chloride were added to the suspension dropwise so that the temperature did not exceed 293 K, then the mixture was stirred continuously for 24 h. The reaction mixture was filtered to remove the unreacted deuterohemine and iron(II) sulfate, the filtrate was diluted with chloroform (50 mL) and washed once in a separating funnel with a 10% ammonia solution and then several times with distilled water. The solvent was evaporated and the porphyrin was exposed to chromatographic purification using Al$_2$O$_3$ (activity grade II) and a mixture of chloroform and methanol (100:1 volume ratio) as the appropriate eluent. The eluate was evaporated to a minimum volume and 13(3),17(3)-dimethyl ester of deuteroporphyrin-IX (Comp. **3**) was precipitated with methanol. The yield was 0.54 g (1.01 mmol), 63%. The spectral characteristics of Comp. **3** are given in the Supplementary Materials (Figures S2 and S3).

13(3),17(3)-*bis*-N-(2-N′,N′-dimethylaminoethyl)amide of deuteroporphyrin-IX (Comp. **4**). A mixture of 7 mL (64.3 mmol) of N,N-dimethylethylenediamine and 0.55 g (1 mmol) of the 13(3),17(3)-dimethyl ester of deuteroporphyrin-IX (Comp. **3**) was dissolved in 10 mL of CHCl$_3$ and was refluxed for 3 h (Scheme), then was diluted with distilled water and neutralized with a dilute solution of AcOH to reach pH = 7. The solvent was evaporated, and then the final compound was purified by column chromatography on silica with a mixture of acetone and CCl$_4$ in the volume ratio of 1:40. The solution was evaporated, and the precipitate was filtered and dried in vacuo. The yield was 0.619 g (0.95 mmol), 93%. The spectral characteristics of Comp. **4** are presented in the Supplementary Materials (Figures S4 and S5).

13(3),17(3)-*bis*-N-(2-N′,N′,N′-trimethylammoniomethyl iodide)amide of deuteroporphyrin-IX (Comp. **5**). A total of 3 mL (48.19 mmol) of methyl iodide was added to a solution of 30 mg (0.05 mmol) of 13(3),17(3)-*bis*-N-(2-N′,N′-dimethylaminoethyl)amide of deuteroporphyrin-IX (Comp. **4**) in 2 mL of CHCl$_3$ (Scheme). The reaction was carried out for 24 h under stirring at room temperature. Chloroform was added to the reaction mixture and the precipitate formed was filtered using a Schott filter. The final compound was washed with CHCl$_3$, then with hexane and dried under vacuo. The yield was 33 mg (0.035 mmol), 76%. The spectral characteristics of Comp. **5** are given in the Supplementary Materials (Figures S6 and S7).

### 2.3. Spectroscopic Measurements

The UV-VIS spectra were registered at room temperature with Drawell D8 and Drawell G9 spectrophotometers (China). The quantitative calculations were based on the linear relationship between the optical density and the PS concentration in the solution according to the Bouguer-Lambert-Beer law. The steady state fluorescence spectra of the preliminarily thermostatted PS solutions were recorded at 298 K with a Solar CM 2203 spectrofluorimeter (Belarus) within the range of 600–750 nm at the excitation wavelength of 405 nm for Comp. **5** and 505 nm in the case of Comp. **6**. The solute concentration was of $7 \times 10^{-6}$ mol kg$^{-1}$ and $1 \times 10^{-6}$ mol kg$^{-1}$ for the spectrophotometric and fluorescence studies, respectively. The spectral characteristics of PSs **5** and **6** are presented in Figure 2 and in the Supplementary Materials (Table S1, Figure S8).

### 2.4. Determination of the Quantum Yield of Singlet Oxygen Generation

The time-resolved photoluminescence of singlet oxygen ($^1$O$_2$) was studied in benzene solutions at 1270 nm with a Lif-200 pulsed laser fluorimeter using a nitrogen laser with the pulse frequency of 30 Hz, the pulse energy of 20 μJ and pulse duration of 2 ns [40]. The quantum yield of $^1$O$_2$ generation ($\gamma_\Delta$) was determined by a comparative method using H$_2$TPP as a suitable standard with the known $\gamma_\Delta$ parameter. The values of $\gamma_\Delta$ were determined by Equation (1):

$$\gamma_\Delta / \gamma_{\Delta\text{st}} = I_0 \times A / I_{0\text{st}} \times A_{\text{st}} \tag{1}$$

where $I_0$ and $I_{0st}$ are the initial $^1O_2$ luminescence intensities of the PS solution studied and the standard determined from the decay curves at $\tau \to 0$; $A$ and $A_{st}$ are the optical densities of the PS and H$_2$TPP solutions. To calculate the $\gamma_\Delta$ quantity of the studied compound in benzene, the value of $\gamma_{\Delta st}$ earlier published for H$_2$TPP and equal to 0.62 was applied [41]. The calculated quantities are listed in Table 1.

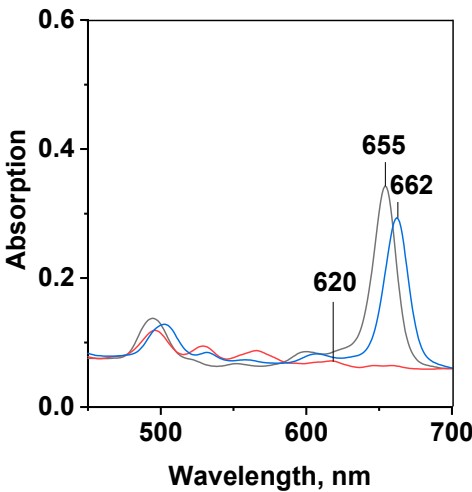

**Figure 2.** UV-VIS spectra of: dicationic porphyrin (Comp. **5**, red line), dicationic chlorin e$_6$ (Comp. **6**, black line) and chlorin e$_6$ trisodium salt (Comp. **7**, blue line) in ethanol at $m_{PS}= 7.3 \times 10^{-6}$ mol/kg.

**Table 1.** Physicochemical characteristics of PSs **5–7**.

| Parameter | Comp. 5 | | Comp. 6 | | Comp. 7 | |
|---|---|---|---|---|---|---|
| | | | Singlet oxygen quantum yield, $\gamma_\Delta$ [a] | | | |
| | - | | 0.60 [b] [14] | | 0.67 [b] [14] | |
| | 0.71 (C$_6$H$_6$) | | - | | 0.61 (C$_6$H$_6$) [14] | |
| | | | Aqueous solubility and partition coefficient | | | |
| $T$, K | Solubility, mg/mL | $P$ | Solubility, mg/mL | $P$ [14] | Solubility, mg/mL | $P$ [19] |
| 298.15 | 0.120 ± 0.017 | 0.25 ± 0.03 | 0.038 ± 0.004 | 1.04 ± 0.02 | >5 | 1.88 ± 0.06 |
| 308.15 | 0.135 ± 0.006 | 0.14 ± 0.07 | 0.046 ± 0.004 | 1.36 ± 0.03 | - | 1.90 ± 0.09 |
| 318.15 | 0.167 ± 0.023 | 0.11 ± 0.02 | 0.054 ± 0.003 | 1.61 ± 0.03 | - | 1.91 ± 0.10 |

[a] The experimental uncertainties are found to be less than 10%; [b] measured in OctOH.

### 2.5. Solubility Study

The solubility of dicationic PSs **5** and **6** was measured by the isothermal saturation method [20,42] with the spectrophotometric control of the solute equilibrium concentration. The data obtained are presented in Table 1.

### 2.6. Determination of Partition Coefficients

The partition coefficients between 1-octanol (OctOH) and phosphate saline buffer (PSB) were determined by the isothermal saturation method described previously [7,13,14,20]. An appropriate amount of a PS solution in OctOH with an initial concentration of 5–80 μmol/kg was placed into a glass cell and the second component of the immiscible solvent system was added in the OctOH-to-PSB volume ratio of 40:60. The solution was subjected to intensive magnetic stirring for up to 36 h until the interface equilibrium was reached. The temperature was maintained during the experiment with an accuracy of ±0.05 K. After 2 h, 2–3 mL of the analyzed phase required for the complete phase separation were withdrawn with a steel needle and weighed. The equilibrium molal concentration of the photosensitizer

was determined spectrophotometrically using the calibration plots obtained earlier. The partition coefficients ($P$) were calculated according to the equation:

$$P = m_{OctOH}/m_{aq} \tag{2}$$

where $m_{OctOH}$ and $m_{aq}$ are the solute equilibrium molalities in OctOH and PSB, respectively. The experimental results are given in Table 1.

### 2.7. Dynamic Light Scattering Measurements

The size of the aggregates formed by PS and their size distribution were studied in aqueous solutions by the dynamic light scattering (DLS) method using a Zetasizer Nano ZS ZEN3600 analyzer (Malvern Instruments, UK) supplied with a $\lambda = 633$ nm laser and the non-invasive backscatter technology (NIBS) of the optical registration system. The scattered light detector was located at an angle of $173^0$ to the incident light. The coefficients of dynamic viscosity and refractive index of the water ($\eta = 0.8872$ mPa/s and n = 1.330) as a dispersion medium were chosen. The PS solutions were prepared by the gravimetric method, filtered with a cellulose filter (Jetbiofil, 0.22 μm) and stored in a dark place for several days before the DLS experiments. The results are presented in Table S2 of the Supplementary Materials.

### 2.8. Spectrophotometric Titration of PS Aqueous Solutions with Tween 80

The stock solutions of porphyrin PS **5** ($m = 7.3 \times 10^{-6}$ mol/kg) and non-ionic surfactant Tween 80 ($m = 7.3 \times 10^{-3}$ mol/kg) were prepared by dissolving an accurate sample of the substances in distilled water. Aqueous solutions with a constant PS concentration and different surfactant concentrations were prepared as probes for spectrophotometric titration by diluting the initial concentrated solution of Tween 80 with water. Each time 1 g of the stock PS solution was placed into the test tube and the calculated amounts of the surfactant and distilled water were added to reach the total mass of the mixture equal to 10 g. The UV-VIS spectra of each solution with a PS/Tween 80 molar ratio varying from 1/5 to 1/150 were recorded at 298 K.

To calculate the value of the equilibrium constant and stoichiometry of the molecular complexation between PS and Tween 80 we applied the two-parameter model proposed recently [16,30,34]. The model parameters given in Table 2 were recovered from the experimental titration curves by fitting them to the equation:

$$\lg\left(\frac{A - A_0}{1 - (A - A_0)}\right) = \lg(K_b) + n \times \lg\left(m_m^T - n \times \frac{m_{PS} \cdot (A - A_0)}{A_{max} - A_0}\right) \tag{3}$$

where $m_{PS}$ is the analytical PS concentration in the solution; $m_m^T = m_T - CMC$ is the concentration of Tween 80 in the micelles, defined as the difference between its analytical concentration and critical micellar concentration ($CMC = 1.5 \times 10^{-5}$ mol/kg); $n$ is the average number of surfactant molecules interacting with a PS unit in a micelle; and $A_0$ and $A_{max}$ are the optical densities of the PS solution in water and the aqueous solution of Tween 80, where the $A$-$f$ ($m_T$) plot reaches a plateau.

Equation (3) can be linearized in the coordinates $\lg([PS \cdot T_n]/[PS]) - \lg[T]$, where the slope angle tangent is parameter n, and the logarithm of the binding constant $\{\lg(K_b)\}$ is the segment cut off on the y-axis. The model parameters $K_b$ and $n$ presented in Table 2 are calculated using the standard iteration method. The parameters of the experimental titration curve are given in the Supplementary Materials (Table S3).

**Table 2.** Parameters of micellar binding (Equation (3)) and localization (Equation (4)) of ionic PSs in aqueous solutions of Tween 80 and KI.

| Parameter | Comp. 5 | Comp. 6 [34] | Comp. 7 [18] |
|:---:|:---:|:---:|:---:|
| | Equation (3) | | |
| $m_{T1}$, mol/kg | $(0.37–1.81)\cdot 10^{-4}$ | $(3.9–14.6)\cdot 10^{-4}$ | $(0.18–1.7)\cdot 10^{-4}$ |
| $m_{T2}$, mol/kg | $(2.35–5.48)\cdot 10^{-4}$ | $(3.1–5.5)\cdot 10^{-3}$ | $(1.7–6.7)\cdot 10^{-4}$ |
| $n_1$ | $1.08 \pm 0.02$ | $1.76 \pm 0.09$ | $0.71 \pm 0.07$ |
| $n_2$ | $2.22 \pm 0.27$ | $3.26 \pm 0.28$ | $2.18 \pm 0.30$ |
| $\log(K_b)_1$ | $4.94 \pm 0.09$ | $5.66 \pm 0.29$ | $3.73 \pm 0.31$ |
| $\log(K_b)_2$ | $9.30 \pm 0.90$ | $9.09 \pm 0.67$ | $9.36 \pm 1.04$ |
| | Equation (4) | | |
| $K_{SV}$, kg/mol | $10.92 \pm 0.30\ (60)$ [1] | $7.78 \pm 0.08\ (60)$ | $2.62 \pm 0.03\ (60)$ |

[1] The values given in parentheses are the Tween 80/PS molar ratio in the solution. The uncertainties represent the standard error.

### 2.9. PS Fluorescence Quenching in Aqueous Solutions of Tween 80 in the Presence of KI

Aqueous solutions of photosensitizers containing a 60-fold molar excess of Tween 80 were titrated with a freshly prepared solution of potassium iodide. Quenching fluorescence studies were performed as follows: a 0.1 g KI solution ($m = 3$ mol/kg) was gradually added to a 2 g PS solution, thermostatted at 298 K, with the spectra recorded at each titration step. The dynamic equilibrium constants ($K_{SV}$) were estimated from the titration curve using the Stern–Volmer equation [34,43]:

$$F_0/F = K_{SV}[I^-] + 1 \qquad (4)$$

where $F_0$ and $F$ are the fluorescence intensities in the aqueous solution of Tween 80 in the absence of the quencher and in the potassium iodide solutions, respectively; and $[I^-]$ is the concentration of the fluorescence quencher. The data are given in Table 2 and in the Supplementary Materials (Table S4).

### 2.10. Preparation of PS Solutions and Bacterial Cultures for the Dark and Photoinduced (aPDT) Antibacterial Activity Studies

Standard microbiological methods [44] were used to study the dark and photoinduced toxicity of the porphyrin dicationic PS **5** towards the following opportunistic pathogens in vitro—*Enterobacter cloacae*, *Escherichia coli* and *Pseudomonas aeruginosa*. All the nosocomial antibiotic-resistant strains of Gram-negative bacteria were carefully grown in the clinical laboratory of the Ivanovo regional clinical hospital. The microbiological studies of the PS cyto- and photocytotoxicity were carried out in certified laboratories of the Department of Microbiology and Research Center of Ivanovo State Medical Academy.

PS solutions were prepared in bidistilled water or in aqueous solutions of Tween 80 or Trilon B ($Na_2H_2Edta$). The final solution was then homogenized on a Sonopuls ultrasonic homogenizer (Bandelin electronic GmbH & Co., Berlin, Germany).

### 2.11. Preparation of the Seed Dose of Test Cultures

Cultures of test strains were grown on beef infusion agar slant in an incubator overnight. The cell density was adjusted to the 5 McFarland standard in sterile saline, which corresponds to approximately $5 \times 10^8$ CFU/mL. The inoculum dose of $2 \times 10^7$ CFU/mL from the stock standard suspension was prepared by diluting. To simulate aPDT, 0.5 mL of the tested microorganisms were added to 4-well plates and incubated in the dark for 0.5 h at room temperature with 0.5 mL of various photosensitizer solutions. After incubation half of the samples were irradiated with red light. Then, all the plates were kept for 24 h at 310 K. To confirm the bactericidal effect in the samples studied, the latter were seeded from all the wells onto Petri dishes with a solid nutrient medium to count the CFU number. The results of the microbiological experiments are presented in Table 3 of the manuscript and in Table S5 of the Supplementary Materials.

**Table 3.** In vitro inactivation of nosocomial Gram-negative pathogens using dicationic Comp. **5** ($m_{PS}$ = 0.1 mmol/kg) and trianionic PS "Fotolon" ($m_{PS}$ = 0.75 mmol/kg).

| | Pathogen, Initial Number of CFU $\times 10^7$ | | |
| --- | --- | --- | --- |
| | *Escherichia coli* | *Enterobacter cloacae* | *Klebsiella pneumonia* |
| | | Darkness | |
| PS + 0.5 wt% Tween 80 | $10^7$ | $10^7$ | 0 |
| PS + 0.1 wt% Na$_2$H$_2$Edta | 5 | 0 | 5 |
| PS + 0.001 wt% H$_2$O$_2$ | $10^7$ | $10^7$ | 2 |
| "Fotolon" [1] | $10^5$ | - | $10^5$ |
| | | Dose 40 J/cm$^2$ | |
| PS + 0.5 wt% Tween 80 | $10^7$ | $10^7$ | 0 |
| PS + 0.1 wt% Na$_2$H$_2$Edta | 5 | 0 | 0 |
| PS + 0.001 wt% H$_2$O$_2$ | 0 | $10^7$ | 0 |
| | | Dose 80 J/cm$^2$ | |
| PS + 0.5 wt% Tween 80 | 0 | $10^7$ | 0 |
| PS + 0.1 wt% Na$_2$H$_2$Edta | 0 | 0 | 0 |
| PS + 0.001 wt% H$_2$O$_2$ | 0 | $3 \cdot 10^6$ | 0 |
| "Fotolon" [1] | $5 \times 10^2$ | - | $10^3$ |

[1] The initial number of CFU and the radiation dose were $10^5$ and 120 J/cm$^2$, respectively.

### 2.12. aPDT Modeling Technique In Vitro

Modeling of microorganism photoinactivation in aqueous solutions was carried out in a dark place at room temperature by irradiating plates using a special LED source of visible light (BMC, Belarus) with adjustable radiation power and water cooling. The maximum radiation power of the LED panel used was ~0.2 W/cm$^2$; the illuminated surface area was up to 100 cm$^2$; the wavelength range of the incident light used for the porphyrin PS was $620 \pm 60$ nm. The radiation power, the sample distance (~10 cm) and the exposure time were selected so as to give a radiation dose of 40 or 80 J/cm$^2$ during a typical therapeutic procedure (~10–15 min) applied in clinics. The radiation dose from the calibrated LED panel was measured with an Argus-03 radiation power meter (Russia).

## 3. Results and Discussion

### 3.1. Physicochemical Studies

The modern requirements for photosensitizers for biomedical applications are versatile and any attempt to synthesize the "ideal drug" is predetermined to failure, since in this case the PS molecule should demonstrate the best qualities not only from the point of view of organic synthesis, photophysics, solution chemistry, cytology, and microbiology, but also from an economic and social standpoint [8,21,29]. Particularly, photosensitizing agents for antimicrobial PDT have to be water-soluble and amphiphilic, to generate reactive oxygen species (ROS), to remain unaggregated in aqueous solutions, and to be selectively accumulated by pathogenic cells of different origins. The most important feature of PS for aPDT is the presence of cationic groups, essentially improving the permeability of the outer membrane of Gram-negative bacteria [3–5,11,29].

#### 3.1.1. Spectroscopic Characteristics of Cationic PSs in Organic Solvents

The location and intensity of the red-shifted $Q_x$(0-0)-band in the absorption spectra and the spectral characteristics of any photosensitizer provide the most important information about light penetration through living tissues and the efficacy of ROS production [4,8–10,12,14,45]. The UV-VIS and fluorescence spectra of Comp. **5** and **6** are typical of porphyrin and chlorin macrocycles, respectively (Figure 2, Table S1 of the Supplementary Materials). The long-wavelength $Q_x$(0-0) and Soret bands of the tetrapyrrolic PSs are induced by a π-π*-electron transfer within the 18π-electronic macroheterocyclic chromophore. The spectra of dicationic porphyrin **5** contain an intense B-band (Soret) at about 400 nm (lg$\varepsilon \approx 5$) and four less pronounced (lg$\varepsilon \approx 4$) Q-bands with the absorption maxima ranging

from 490 to 620 nm. Hydration of the π-bond in one of the pyrrole rings (17,18-positions) of the macrocycle results in the molecule polarization and shifts the Q-band to 660 nm with an intensity increase (lg$\varepsilon \approx 4.5$) in the long-wavelength absorption within the so-called optical window of tissue of dicationic chlorin **6**. The same type of the UV-VIS-spectra is characteristic of chlorin e$_6$ (Comp. **7**), the active compound of many clinically approved PSs [5–7]. The bands in the fluorescence spectra of compounds **5** and **6** are characterized by moderate Stokes shifts ranging from 2 to 11 nm (see Table S1 in the Supplementary Materials). This behavior is known to be typical of aromatic macroheterocycles with a relatively planar structure [46,47].

The direct photoluminescent method was used to estimate the quantum yield of the singlet oxygen generation ($\gamma_\Delta$) by dicationic PSs in apolar non-aqueous media (Table 1), modeling the pseudo-lipid environment of the dye in biomembranes [48–50]. Regardless of the tetrapyrrolic macrocycle type and non-polar solvent used [14] as well as the type and number of charged groups in a PS molecule, the $\gamma_\Delta$ value measured in benzene and 1-octanol was found to be sufficiently high and ranged from 0.60 for dicationic chlorin **6** to 0.71 for dicationic porphyrin **5**. The quantity determined for the porphyrin PS was slightly higher than those of the naturally derived chlorins and, specifically, chlorin e$_6$ (Comp. **7**). This situation is consistent with the data presented earlier in the literature, although the extinction coefficient $\varepsilon$ and, as a result, photon absorption probability in the long-wavelength Q$_x$(0-0)-band of the porphyrin PS are noticeably lower [8–10].

An analysis of the photophysical parameters reveals that both dicationic compounds can be considered good candidates for photodynamic inactivation of pathogens.

### 3.1.2. Solvation, Partition and Aggregation of PSs in Aqueous Solutions

One of the main requirements for medical preparations as a whole and photosensitizers in particular is solubility in water or aqueous solutions of biocompatible carriers suitable for reaching the drug therapeutic concentration [22,48,51–53]. Therefore, the water solubility of dicationic PSs **5** and **6** within 298–318 K was measured (Table 1). The solubility of both macroheterocycles is high enough to reach the therapeutic concentrations used in clinics [5–8,20]. The results presented in Figure 3 and in Table 1 clearly demonstrate that the solubility of porphyrin PS is several times higher than that of the chlorin derivative wherein, as expected, this quantity increases with the temperature for both compounds investigated (Figure 3). The higher solubility of porphyrin PS **5** is consistent with the decreasing value of the partition coefficients $P$ measured for this dicationic compound in the OctOH–PSB immiscible solvent system. $P$ value of porphyrin **5** is notably lower compared to both chlorin dyes carrying two or three charged groups (Table 1).

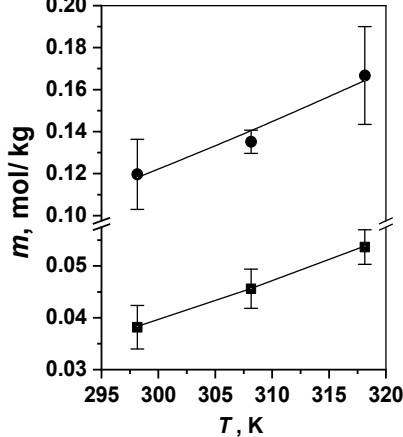

**Figure 3.** Temperature dependences of the dicationic porphyrin (Comp. **5**, ●) and chlorin (Comp. **6**, ■) solubility in water at 298–318 K.

The PS partition provides important information about the affinity of the solute towards similar lipid-like and water-like phase modeling water–lipid interfaces in biosystems [18,48–50,54,55]. The ionic groups of charged molecules **5**–**7** are responsible for the pronounced affinity of the macroheterocyclic molecules to the water-like compartments, which increases as follows in the series of PSs: **7** < **6** < **5** (Table 1). Indeed, if the chlorin PS (Comp. **6**) carrying two cationic groups in the tetrapyrrole framework is almost equally distributed between 1-octanol and water, then only one fifth of the most hydrophilic dicationic porphyrin **5** is accumulated in the OctOH phase (Table 1). Thus, the amphiphilic water-soluble chlorin $e_6$ is found to be the most "hydrophobic" of the PSs investigated. The $P$ values appear to be almost temperature-independent for triply charged chlorin $e_6$ (Comp. **7**) [14,19] in opposition to the porphyrin PS **5**. This finding leads to the assumption about PS **5** being more prone to aggregation in aqueous solutions. At the same time, dicationic chlorin PS **6** demonstrates only moderate T dependence of the partition coefficient.

The UV-VIS spectra of macrocycles **5** and **6** measured in organic solvents of different polarity, such as DMF, ethanol or benzene, demonstrate similar absorption of non-aggregated molecules [33]. The dissolution of cationic PSs in water broadens and shifts the absorption bands, as well as quenches fluorescence, indicating the aggregation of the pigments in aqueous solutions [56,57]. In some cases, the aggregates formed can be indistinguishable on the photon correlation spectroscopy scale, if they are nano-sized and presented by PS dimers or trimers [57,58].

The nanoaggregate formation depends strongly on the PS structure. Our dynamic light scattering (DLS) studies show that, despite its good water solubility, dicationic porphyrin **5** demonstrates the formation of nanoaggregates with a hydrodynamic diameter of about 350 nm, even at a very low concentration of $7 \times 10^{-6}$ mol/kg. Starting from the PS concentration of about 0.2 mmol/kg, large aggregates of 2000–3000 nm are observed (Figure 4). On the contrary, chlorin **6** is found to be nanoaggregated in aqueous solutions only at millimolar concentrations (200 nm at 0.73 mmol/kg), while in the micromolar range, this PS exists in the form of small (<1 nm) aggregates not observed by the DLS techniques but registered by UV-VIS and fluorescent spectroscopy. Trianionic chlorin $e_6$ (Comp. **7**) does not form nanoaggregates in water even in the "therapeutic" millimolar concentration range [18].

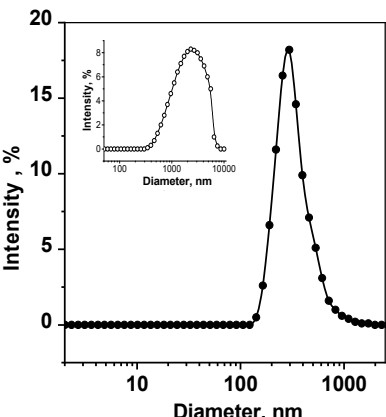

**Figure 4.** Size distribution of Comp. **5** aggregates in an aqueous solution; $m_{PS}$ equals 0.0017 mmol/kg for the main picture and 0.730 mmol/kg for the insert.

Aggregation of PS molecules in aqueous solutions is known to significantly reduce their ability to generate singlet oxygen, their affinity to the cell membranes and membrane transport and, consequently, to minimize the PDT efficacy of pathogen photoinactivation as a whole. Since the dicationic porphyrin PS **6** revealing high water solubility is, nevertheless, prone to aggregation at low concentrations, it is required to apply an appropriate carrier to prevent this kind of hydrophobic interaction [18,22,30,51–53,57–59].

To avoid the aggregation is one of the most important tasks in the development of a new generation of PSs. In this regard, our next study is aimed at studying the mechanism of interaction of charged photosensitizer molecules with a potential nanocarrier–biocompatible non-ionic micellar surfactant Tween 80 [30,60,61].

### 3.1.3. PS interaction with Micellar Surfactant Tween 80

The spectrophotometric and fluorescent titrations were carried out to investigate the PS binding to Tween 80 and to establish the location of the dye molecules in the surfactant micelle (Table 2, Figure 5). The data on the UV-VIS and fluorescence spectra presented in Table S1 of the Supplementary Materials clearly demonstrate the disaggregation of the PSs in the aqueous solutions containing small amounts of Tween 80. The disaggregation is accompanied by both a fluorescence build-up and an intensity increase in the absorption spectra [18,20,22,30,33], which is confirmed by the DLS results [57,58].

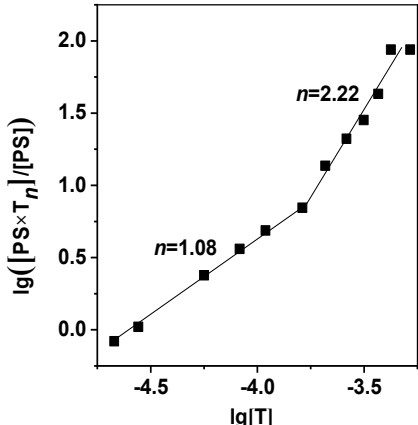

**Figure 5.** Curves of $\lg([PS \cdot T_n]/[PS])$ vs. equilibrium molality of Tween 80 for Comp. **5**. The dots represent the experimental data and the solid lines—the values calculated by Equation (3).

All the three charged photosensitizers (Comp. **5**–**7**) are found to form relatively stable molecular complexes with a passive carrier demonstrating two modes of PS–Tween 80 binding with their own stability constant and stoichiometry of the interaction between the macrocycle and surfactant molecules (Table 2, Figure 5). The first stage of PS–Tween 80 binding is described by the value of $\lg(K_b)_1$, ranging from 4 to 6, suggesting quite strong molecular complexation. In the range of low surfactant concentrations and micellar deficiency, the photosensitizer, according to the n value (Table 2, Equation (3)), interacts with one or two Tween 80 molecules. An increase in the number of micelles in the aqueous solution results in more feasible PS contact with a larger number of Tween 80 molecules, to the higher $K_b$ and $n$ values. However, this regularity is more concerned with the PS molecules localized in the outer layer of the surfactant micelle.

To gain a deeper insight into the mechanisms of PS–Tween 80 interaction, we studied the dye fluorescence quenching in aqueous solutions of potassium iodide containing a certain amount of Tween 80. The quenching curves in all the cases are described quite well by the Stern–Volmer Equation (4). The $K_{SV}$ values determined from the dependences $F_0/F\text{-}f$ ($[I^-]$) are calculated at the PS–Tween 80 molar ratio of 1:60. An excess of Tween 80 is related to the aggregation number of this surfactant in water [29,58]. Both $K_{SV}$ values calculated for dicationic PSs **5** and **6** are quite high, indicating that the dye molecules are located in the outer layer of the micelle, surrounded by polar oxyethylene groups, where the hydrated quenching ions can easily penetrate. Chlorin $e_6$ (Comp. **7**), bearing three anionic groups, demonstrated different behavior in the fluorescence quenching process with the Stern–Volmer constant being at least 50% lower than that of the dicationic dyes under similar conditions (Table 2). Presumably, the photosensitizers containing negatively

charged groups were more deeply incorporated into the Tween 80 micellar structure than similar cationic dyes.

Thus, the results of the spectroscopic studies performed in aqueous solutions indicate strong binding of macroheterocycles **5** and **6** to Tween 80 micelles, which can be considered convenient vehicles of passive PS transport to the surface of the living cells.

### 3.2. Photoinduced Inactivation of Antibiotic-Resistant Bacterial Microflora

Clinically approved macroheterocyclic photosensitizers carrying anionic functional groups are known to have low affinity to the outer lipopolysaccharide membrane of Gram-negative bacteria, poorly penetrate inside the pathogen cell and, therefore, demonstrate only moderate photoinduced antimicrobial activity [3–5,11,13–15,44]. For instance, the anionic drug "Fotolon" based on chlorin $e_6$ trisodium salt (Comp. **7**) and polyvinylpyrrolidone (PVP) as an appropriate passive carrier inactivates in vitro the Gram-positive bacteria *Staphylococcus aureus* almost completely, but demonstrates only 1–2 logs of killing for Gram-negative pathogens [15,44,62,63]. The pronounced versatility of photosensitizers substituted by cationic groups at photoinactivation of the opportunistic microorganisms is the main reason for the profound interest to these compounds [2–5,13,29].

Our recent preliminarily studies [14,33] of the photoinactivation of the archival pathogens *Staphylococcus aureus, Escherichia coli* and *Candida albicans* with dicationic PSs **5** and **6** have demonstrated a higher antimicrobial potential of the deuteroporphyrin-IX derivative. Thus, here we present our experimental results (Table 3) of applying porphyrin PS **5** in in vitro photoinactivation of the hospital strains of *Escherichia coli*, *Enterobacter cloacae* and *Klebsiella pneumonia,* resistant to a series of antibiotics—"Ciprofloxacin", "Gentamicin sulfate", "Ceftriaxone", "Nevigramon" (Nalidixic acid), "Furadonin" (except *E. coli*), etc. The data presented in Table 3 illustrate several important issues: firstly, expected higher efficacy of the cationic PS **5** in comparison with the anionic "Fotolon"; secondly, a pronounced light-dose-dependent killing effect of Gram-negative pathogens and, more importantly, aPDT potentiation effect caused by adding appropriate carriers, such as Tween 80 [15,60], or other potentiation additives enhancing the permeability of the outer bacterial membranes ($Na_2H_2Edta$) [44] or changing the mechanisms of ROS generation ($H_2O_2$) [12,44,64,65].

As a rule, small amount (0.1 wt%) of $Na_2H_2Edta$ are low-toxic for pathogens [4,15,44]. However, the addition of such a Trilon B quantity in the presence of the Comp. **5** leads to pronounced dark toxicity (Table 3), indicating free PS penetration through the outer membrane of all the three pathogens. Additions of Tween 80 and $H_2O_2$, on the opposite, make it possible to observe photoinactivation of *Escherichia coli* instead of the dark toxicity.

It has been established that a 40 J/cm$^2$ dose of light in Tween 80 solutions is not enough to kill *Escherichia coli*, while 80 J/cm$^2$ irradiation guarantees its complete inactivation (Table 3).

In the case of *Enterobacter cloacae* and *Klebsiella pneumonia*, no pure photodynamic inactivation of the pathogens is observed. Enterobacteria are resistant in all cases, with the exception, as already noted, of small additions of $Na_2H_2Edta$. In turn, *Klebsiella pneumonia* almost completely dies when incubated in the dark, and irradiation with a 40 J/cm$^2$ dose only guarantees the absence of residual pathogen growth.

Notably, the use of the second-generation anionic PS "Fotolon", despite the higher concentration and dose of light used, is found to be a significantly less effective photosensitizing agent as compared to cationic porphyrin **5** (Table 3).

### 4. Conclusions

In conclusion, the dicationic deuteroporphyrin-IX derivative synthesized and presented here as a photosensitizing agent (PS) for antibacterial PDT demonstrates a good quantum yield of singlet oxygen, good water solubility ranging at the therapeutic concentration, appropriate amphiphilicity and ability to be solubilized due to the molecular complexation with the micellar surfactant Tween 80. Despite the higher cationic porphyrin solubility in water compared to the chlorin of a similar structure it is found to be strongly

aggregated in an aqueous environment even at micromolar concentrations. Therefore, it is strongly recommended to use passive delivery vehicles for efficient PS solubilization in water. Aggregation of the porphyrin PS studied is easily prevented using small amounts of a solubilizing agent, such as biocompatible surfactant Tween 80, to form stable PS–Tween 80 complexes carrying the photosensitizer localized at the micelle surface near the hydrophilic head-groups. Dicationic porphyrin demonstrates a pronounced photoinduced antimicrobial effect towards nosocomial antibiotic-resistant Gram-negative bacterial strains, especially in the presence of potentiating agents, such as Trilon B, Tween 80 or $H_2O_2$.

**Supplementary Materials:** Ref. [66] is cited in Supplementary Materials. The following supporting information can be downloaded at: https://www.mdpi.com/article/10.3390/photochem3010011/s1, Figure S1: synthesis of 13(3),17(3)-bis-N-(2-N′,N′,N′-trimethylammonioethyliodide) amide of deuteroporphyrin-IX (Comp. 5) from hemin (Comp. 1); Figure S2: 1H NMR spectrum of 13(3),17(3)-dimethyl ester of deuteroporphyrin-IX (Comp. 3) in CDCl3; Figure S3: Mass spectrum (MALDI-TOF) of 13(3),17(3)-dimethyl ester of deuteroporphyrin-IX (Comp. 3). DHB was used as a matrix; Figure S4: 1H NMR spectrum of 13(3),17(3)-bis-N-(2-N′,N′-dimethylaminoethyl) amide of deuteroporphyrin-IX (Comp. 4) in CDCl3; Figure S5: Mass spectrum (MALDI-TOF) of 13(3),17(3)-bis-N-(2-N′,N′-dimethylaminoethyl) amide of deuteroporphyrin-IX (Comp. 4). DHB was used as a matrix; Figure S6: 1H NMR spectrum of 13(3),17(3)-bis-N-(2-N′,N′,N′-trimethylammoniomethyl iodide) amide of deuteroporphyrin-IX (Comp. 5) in CD3OD; Figure S7: Mass spectrum (MALDI-TOF) of 13(3),17(3)-bis-N-(2-N′,N′,N′-trimethylammoniomethyl iodide) amide of deuteroporphyrin-IX (Comp. 5). DHB was used as a matrix; Figure S8: The UV-VIS spectra of Comp. 5 (mPS = $7.3 \times 10-5$ mol/kg) in water and aqueous solutions of Tween 80 at room temperature. Curves show a gradual increase in the Tween 80 concentration: (■)-0, (□)-$7.3 \times 10-4$ (●)-$1.5 \times 10-3$ and (○)-$1.3 \times 10-2$ mol/kg, respectively. Inset: dependence of the optical density of the Comp. 5 solution on the surfactant concentration at $\lambda = 410$ nm. Table S1: parameters of the absorption and fluorescence spectra for Comps. 5, 6; Table S2: size distribution of aggregates in aqueous solution: Comp. 5, mPS = 0.0175 mmol/kg, Comp.6, mPS = 0.789 mmol/kg; Table S3: experimental values of the titration curves of Comp. 5 aqueous solutions (m = $7.3 \times 10-6$ mol/kg) with Tween 80 solution; Table S4: dependence of PS 5 fluorescence intensity in Tween 80 solutions on potassium iodide concentration; Table S5: in vitro inactivation of nosocomial Gram-negative pathogens (mPS = 0.0001 mol/kg).

**Author Contributions:** Conceptualization, A.V.K. and D.B.B.; methodology, A.V.K., E.A.V. and D.B.B.; software, M.O.K.; validation, N.V.K. and D.B.B.; formal analysis, S.O.K.; investigation, S.O.K., E.A.V. and A.V.K.; resources, M.O.K. and D.B.B.; data curation, A.V.K. and D.B.B.; writing—original draft preparation, N.V.K. and D.B.B.; writing—review and editing, D.B.B.; visualization, N.V.K. and S.O.K.; supervision, A.V.K.; project administration, D.B.B.; funding acquisition, D.B.B. and A.V.K. All authors have read and agreed to the published version of the manuscript.

**Funding:** This research was funded by the Russian Science Foundation (project No. 21-13-00398; https://rscf.ru/project/21-13-00398, accessed on 20 April 2021).

**Data Availability Statement:** The data presented in this study are available on request from the corresponding author.

**Acknowledgments:** We are grateful to the Centers for the Shared Use of Scientific Equipment of ISUCT (support of the Ministry of Science and Higher Education of Russia, grant No. 075-15-2021-671) and ISC RAS. We also thank Garas'ko E.V. (Ivanovo State Medical Academy) for her invaluable help in providing microbiological studies.

**Conflicts of Interest:** The authors declare no conflict of interest.

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
