# Peer review of "Water-Soluble Dicationic Deuteroporphyrin Derivative for Antimicrobial PDT: Singlet Oxygen Generation, Passive Carrier Interaction and Nosocomial Bacterial Strains Photoinactivation"

_2673-7256, doi:10.3390/photochem3010011_

Round 1

Reviewer 1 Report

A cationic porphyrin was synthesized, which has better solubility in water than 17(3)-dimethyl ester of chlorin e6(Comp. 6) and chlorin e6 trisodium salt (Comp. 7). It reacts with inert nonionic carrier Tween 80 to form stable nano-compounds. Moreover, the compounds produce singlet oxygen in nonpolar solvents and have good antibacterial properties. However, there are some problems in this article, which need to be answered and revised by the author.

1.     Is there no solubility data for Compound 7 in Table 1 because it is insoluble in water? If the compound can be dissolved in water, please supplement its solubility data.

2.     The singlet oxygen yields of the same compound in different solvents are different. Compound 5, Compound 6 and Compound 7 are determined in benzene and octanol respectively, so the comparison is of little significance. The authors can choose the same solvent, and then determine the singlet oxygen yield of the three compounds.

3.     Compound 5 and compound 6 in Fig. 2 should be distinguished by different colors, and the UV absorption spectrum data of compound 7 is missing.

4.     On page 9, line 356, the author stated that the P value of compound 6 has no temperature dependence. But based on the information in Table 1, I come to the opposite conclusion that the P value of compound 6 increases with the increase of temperature, and there is an obvious temperature dependence. So I need an explanation from the author.

5.     In photodynamic antimicrobial experiment, compound 5 was selected and added with 0.5% wt Tween 80, 0.1% wt Na2H2Edta and 0.001% wt H2O2 respectively. In this experiment, I have a question, does compound 5 alone have photodynamic antibacterial activity? I need the author to provide this test data.

Author Response

Dear the Reviewer,

Thank you very much for your valuable analysis of our physical chemical and microbiological efforts. Your important questions/comments and our answers are listed below.

Sincerely yours, Dmitry Berezin

Reviewer 2 Report

The manuscript describes preparation of dicationic derivative of deuteroporphyrin, its photophysico-chemical properties, aggregation behaviour and in vitro antimicrobial activity against three antibiotic-resistant strains of opportunistic Gram-negative bacteria. This topic is very relevant and deals with one of the very promising approaches against antimicrobial-resistant pathogens, photodynamic inactivation, and the applications of water soluble, cationic porphyrins, especially in case of Gram-negative bacteria. Also, the new porphyrin was compared with dicationic chlorin, derivative of chlorin e6, which in previous studies was prepared by the same group and which also has two trimethylammonioalkyl iodide groups, and with trianionic chlorin e6. I believe that researchers with similar interests will find these results, especially aggregation studies, useful.

The manuscript is well written and properly organized - the introduction refers to the relevant previous research and the experiments are described in the materials and methods section in sufficient details. Synthesis scheme, synthetic procedures and microbiology protocols are actually described both in the manuscript and in the supplement, which I find redundant. The results are mostly clearly presented (however, data in tables could be more clearly organized) and explained, with appropriate discussion and conclusions.

Minor corrections to be made:

Line 75, Scheme - there are ethyl groups in the structure 1 instead of vinyl groups; drawing of the structure 3 should be corrected (where the ester group is attached to pyrrole on the left).

Line 267 – there is an empty bracket.

Line 315 and line 400 – not to confuse, better say Table S1 (not Table 1).

In supplementary – typo in the title (Deuteroporphyrin).

Figure S1 – there are ethyl groups in the structure 1 instead of vinyl groups; drawing of the structure 3 should be adjusted (where the ester group is attached to pyrrole on the left).

Figure S2. – please comment peaks in the 1H spectrum from around 0.9 – 1.5 ppm (solvent, impurities?). Also in Figure S4.

Please show enlarged parts of the NMR spectra where multiple peaks overlap.

Stokes shifts in Table 1 are given in Dl, not Dn.

Author Response

Dear the Reviewer,

Thank You very much for your evaluation of our modest physical chemical and microbiological efforts. We are grateful for the comments made and tried to take them under consideration. Please see the attachment.

Sincerely yours, Dmitry Berezin
